# Novel anti-PTEN C2 domain monoclonal antibodies to analyse the expression and function of PTEN isoform variants

**Leire Torices**[1], **Caroline E. Nunes-Xavier**[1,2], **José I. López**[1], **Rafael Pulido**[1,3]*

**1** Biobizkaia Health Research Institute, Barakaldo, Spain, **2** Institute for Cancer Research, Oslo University Hospital, Oslo, Norway, **3** Ikerbasque, The Basque Foundation for Science, Bilbao, Spain

* rpulidomurillo@gmail.com, rafael.pulidomurillo@osakidetza.eus

**Data Availability Statement:** All relevant data are within the paper and its Supporting Information files.

## Abstract

*PTEN* is a major tumor suppressor gene frequently mutated in human tumors, and germline *PTEN* gene mutations are the molecular diagnostic of PTEN Hamartoma Tumor Syndrome (PHTS), a heterogeneous disorder that manifests with multiple hamartomas, cancer predisposition, and neurodevelopmental alterations. A diversity of translational and splicing PTEN isoforms exist, as well as PTEN C-terminal truncated variants generated by disease-associated nonsense mutations. However, most of the available anti-PTEN monoclonal antibodies (mAb) recognize epitopes at the PTEN C-terminal tail, which may introduce a bias in the analysis of the expression of PTEN isoforms and variants. We here describe the generation and precise characterization of anti-PTEN mAb recognizing the PTEN C2-domain, and their use to monitor the expression and function of PTEN isoforms and PTEN missense and nonsense mutations associated to disease. These anti-PTEN C2 domain mAb are suitable to study the pathogenicity of PTEN C-terminal truncations that retain stability and function but have lost the PTEN C-terminal epitopes. The use of well-defined anti-PTEN mAb recognizing distinct PTEN regions, as the ones here described, will help to understand the deleterious effects of specific *PTEN* mutations in human disease.

## Introduction

PTEN is a ubiquitously expressed multifaceted tumor suppressor protein that exerts homeostatic functions in all human tissues, mainly by antagonizing the pro-survival PI3K/AKT/mTOR signalling pathway through its phosphatidylinositol 3,4,5-trisphosphate (PIP3) phosphatase activity [1–4]. *PTEN* gene is frequently targeted by mutations in sporadic human cancers, as well as in the germline of individuals with tumor syndromes and/or autism-related neurodevelopmental alterations (PTEN Hamartoma Tumor Syndrome, PHTS, [MIM# 158350]; Macrocephaly/Autism Syndrome, [MIM# 605309]) [5, 6]. Most of *PTEN* mutations found in association with disease are total or partial loss-of-function mutations, either by directly decreasing PTEN catalysis or by diminishing PTEN protein stability. In addition, a variety of *PTEN* disease-associated mutations affect PTEN subcellular distribution in non-

**Funding:** This work has been supported in part by grants BBH-19-001 (to RP) from PTEN Research Foundation (United Kingdom), and SAF2016-79847-R (to RP and JIL), from Ministerio de Economía y Competitividad (Spain and The European Regional Development Fund). LT has been the recipient of a predoctoral fellowship from Asociación Española Contra el Cáncer (AECC, Junta Provincial de Bizkaia, Spain). CN-X is the recipient of a Miguel Servet Research Contract from Instituto de Salud Carlos III (grant number CP20/00008). the funders had no role in study design, data collection and analysis, decision to publish, or preparation of the manuscript.

**Competing interests:** The authors have declared that no competing interests exist.

nuclear and nuclear compartments [7–11]. Nonsense mutations generating premature termination codons (PTC) in the PTEN coding sequence are frequently found in tumors and in the germline of PHTS patients [12]. PTEN proteoforms generated by PTC are in most cases unstable truncated proteins with compromised functional activity, although when PTC are at the PTEN C-terminal region PTEN protein stability and enzymatic function is preserved at larger extent [13, 14].

The PTEN canonical protein contains 403 amino acids, distributed in a N-terminal protein tyrosine phosphatase (PTP) catalytic domain followed by a membrane binding C2 domain and an unstructured regulatory C-terminal tail. This canonical PTEN form is the most abundant PTEN protein, but alternative initiation of translation of the PTEN mRNA generates several PTEN long isoforms containing N-terminal extensions. Translation of PTEN long isoforms initiates with Leu or Ile residues, and its physiologic regulation is mostly unknown [15–17]. Reported PTEN long isoforms include PTEN-L/α (576 amino acids), PTEN-M/β (549 amino acids), and PTEN-O/ε (475 amino acids). PTEN-L/-M/-O isoforms have an intact PTP catalytic domain and display distinct subcellular localization and specific functional properties, likely accounting for differential contribution to physiologic and pathogenic processes [18–26]. In this regard, oncogenic functions have also been proposed for PTEN-L/-M isoforms [27, 28]. In addition, alternative splicing of the precursor PTEN mRNA has also been reported, both under pathogenic and non-pathogenic conditions, and a PTEN isoform lacking the PTEN-C-terminal region encoded in exon 9 (PTEN-Δ, residues 1-343-Ser) has been proposed to play active roles in the context of tumor suppression [29–32].

The detection of PTEN protein using specific anti-PTEN monoclonal antibodies (mAb) is important in diagnostic and prognostic protocols in clinical oncology, including the monitoring of biological samples from PHTS patients. In addition, precise anti-PTEN mAb are essential for the progress of PTEN research in experimental settings. Several well-characterized anti-PTEN mAb are available which recognize the PTEN C-terminus, in part due to the high immunogenicity of this PTEN region. These mAb are highly valuable reagents in the clinics and in research, but most of them do not recognize PTEN isoforms or disease-associated PTEN variants lacking the PTEN C-terminal sequence [33–37]. This constitutes a limitation in view of the variety of PTEN proteoforms that may exist under physiologic and pathogenic conditions. Here, we describe the generation and precise characterization of novel anti-PTEN C2 domain mAb, and illustrate and discuss their use to study the expression and function of PTEN isoforms and disease-associated C-terminal truncated PTEN variants.

## Materials and methods

### Generation and purification of monoclonal antibodies

To obtain the novel anti-PTEN C2 domain mAb-secreting hybridoma (BA226 mAb), Balb/c mice were immunized with the PTEN peptide C<u>SSNSGPTRRREDKFM</u> (226–239 PTEN residues, underlined) conjugated to keyhole limpet hemocyanin, and spleen cells were fused with myeloma SP2/0 cells following standard procedures. Screening of positive hybridoma clones was performed by enzyme-linked immunosorbent assay (ELISA), using the immunogen peptide (1 μg/ml) bound to plastic as the antigen and hybridoma culture supernatant as the source of mAb. Anti-PTEN BA226 mAb (IgG1, k; original fusion clone 18E5-1) was purified from hybridoma culture supernatant using protein A from *S. aureus*, and it was kept in phosphate buffered saline pH 7.4 (PBS). Obtaining and purification of BA226 was custom-made at Abyntek Biopharma. The research was approved by the animal research ethics committee OLAW [ID number: F16-00214 (A5892-01). Valid until: 11-09-2025]. The obtaining of 425A mAb (IgG3, k) has been previously described [33].

## Cell culture, transfections, plasmids, and mutagenesis

Simian kidney COS-7 cells (ATCC CRL-1651) were grown at 37° C, 5% $CO_2$ in DMEM containing high glucose supplemented with 5% heat-inactivated fetal bovine serum (FBS), 1 mM L-glutamine, 100 U/ml penicillin, and 0.1 mg/ml streptomycin. Cells were transfected using GenJet reagent (Signa-Gen) according to the manufacturer instructions, and processed for analysis 48 h after transfection. The pRK5 PTEN, pRK5 GST-PTEN (N-terminal tagging), pRK5 PTEN-L (Met), and pSG5-AKT1 plasmids have been described [38, 39]. pRK5 PTEN-L (Leu), pRK5 PTEN-M (Met), pRK5 PTEN-M (Ile), pRK5 PTEN-O (Met), and pRK5 PTEN-O (Leu) were made by PCR oligonucleotide site-directed mutagenesis from pRK5 PTEN-L (Met), as described [38]. Plasmid pRK5 PTEN-Δ (residues 1-343-Ser) was made by PCR mutagenesis from pRK5 PTEN. The PTEN and GST-PTEN amino acid substitution and truncation variants were made by PCR mutagenesis, and mutations were confirmed by DNA sequencing. Nucleotide and amino acid numbering for PTEN variants correspond to reference sequences from accession numbers NM_000314 and NP_000305, respectively.

## Immunoblotting

Whole cell protein extracts from COS-7 cells transfected with pRK5 plasmids encoding PTEN isoforms or variants (or co-transfected with pSG5 HA-AKT1) were prepared by cell lysis in ice-cold M-PER™ lysis buffer (ThermoFisher Scientific) supplemented with PhosSTOP phosphatase inhibitor and cOmplete protease inhibitor cocktails (Roche), followed by centrifugation at 15200 g for 10 min and collection of the supernatant. Proteins (50–100 μg) were resolved in 10% SDS-PAGE under reducing conditions and transferred to PVDF membranes. Immunoblotting was performed using anti-PTEN BA226, 425A [33], and 6H2.1 (Merck Millipore) [35] mAb, anti-phospho-Ser473-AKT and anti-AKT (Cell Signaling Technologies, USA) antibodies, anti-GST antibody [40], or anti-GAPDH antibody (6C5, Santa Cruz, USA) diluted in immunoblot blocking buffer [*Odyssey Blocking Buffer* (*OBB buffer*, LI-COR Biosciences, USA); diluted 1:1 in PBS], followed by IRDye-conjugated anti-rabbit or anti-mouse antibodies (LI-COR). Determination of reactivity of anti-PTEN BA226 and 425A mAb with PTEN variants was made from at least two independent experiments with similar results. PTEN truncation and amino acid substitution variants were run in parallel with PTEN wild type, and reactivity was determined as similar to PTEN wild type (+), diminished reactivity (+/-), or lack of reactivity (-), taking into consideration the expression levels of each variant as indicated by a reference mAb. For determination of phospho-AKT content, bands were quantified using an Image studio™ software with Odyssey® CLx Imaging System (LI-COR Biosciences).

## Immunofluorescence and immunohistochemistry

COS-7 cells were transfected with pRK5 plasmid encoding PTEN and processed for immunofluorescence as previously described [41], using anti-PTEN BA226 mAb (25 μg/mL) and fluorescein-conjugated anti-mouse antibody. Nuclei were identified by Hoechst (Sigma-Aldrich) staining. Immunohistochemistry was performed in an automated immunostainer (EnVision™ FLEX, Dako Autostainer Plus; Dako), using formalin-fixed, paraffin embedded (FFPE) prostate adenocarcinoma tissue sections, as described [41]. Anti-PTEN BA226 and 6H2.1 mAb were used at 10 μg/ml, and antigen retrieval was performed in Dako PT Link instrument in Tris/EDTA buffer pH 9 (EnVision™ Target Retrieval Solution, High pH).

## Statistical analysis

Statistical significance analysis was performed by the Student's *t* test, and *P* values were calculated for the differences in phospho-AKT/AKT ratio in the presence of PTEN isoforms or variants with respect to the presence PTEN wild type.

## Results

### Characterization of anti-PTEN C2 domain mAb

Anti-PTEN mAb recognizing well-defined epitopes in PTEN protein are valuable reagents in research and in the clinical practice. However, most of the available anti-PTEN mAb recognize epitopes at the PTEN C-terminus, which constitutes a handicap to study the expression of specific PTEN proteoforms [34]. We have previously described an anti-PTEN C2 domain mAb (425A mAb) [33, 41]. Here, we have generated a novel anti-PTEN C2 domain mAb (BA226) by immunizing mice with a PTEN C2 domain peptide encompassing residues 226–239, and we have characterized both BA226 and 425A mAb in terms of precise epitope mapping using recombinant PTEN variants ectopically expressed in mammalian cells. The novel anti-PTEN BA226 mAb is suitable for ELISA, immunoblot, immunofluorescence, and immunohistochemistry techniques (**Fig 1A–1E**). Immunoblot analyses of the reactivity of BA226 and 425A mAb with PTEN N- and C-terminal truncations [made in the background of a fusion protein GST-PTEN (GST N-terminal)] (**Fig 2A and 2B**), and Ala-substitutions (**Table 1**), mapped the BA226 epitope at PTEN residues 230–238, whereas the 425A epitope was mapped at residues 274–287 (**Fig 2A and 2C**; **Table 2**). The localization of the BA226 and 425A epitopes on PTEN 3D structure is shown in **Fig 3**. BA226 epitope locates at the exposed cβ3' sheet, whereas 425A epitope locates at the cα1 helix in the vicinity of the C2 domain-unstructured loop. Several amino acid substitutions generated from *PTEN* gene disease-associated mutations targeting these regions specifically affected the reactivity of each mAb towards PTEN (**Fig 2D and 2E**; **Tables 3 and 4**), which could have clinical use in specific groups of patients. A schematic depiction of the localization of the epitopes recognized by the BA226 and 425A anti PTEN C2 domain mAb, together with that recognized by the 6H2.1 anti-PTEN C-terminus mAb [34], is also shown in **Fig 4A**.

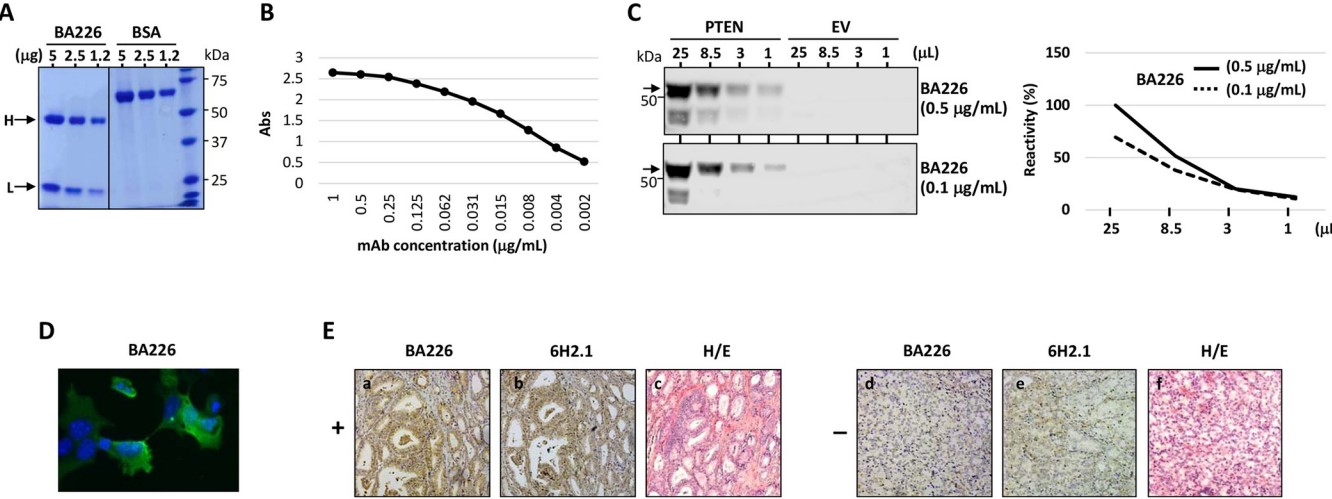

**Fig 1. Characterization of anti-PTEN BA226 mAb reactivity.** (**A**) Coomassie staining of purified BA226 mAb, resolved in 10% SDS-PAGE gel under reducing conditions (H, immunoglobulin heavy chain; L, immunoglobulin light chain). Bovine serum albumin (BSA) is shown as a comparison. (**B**) ELISA determination of binding of BA226 mAb to the PTEN peptide used for immunization. Decreasing concentrations of mAb were used, as indicated. (**C**) Recognition of PTEN by BA226 mAb, as monitored by immunoblot. Decreasing amounts of lysates (0.8 µg/µl) from COS-7 cells transfected with pRK5 plasmid encoding human PTEN or with empty vector (EV) were subjected to immunoblot using different concentrations of BA226 mAb, as indicated. The arrow indicates the migration of recombinant PTEN. In the right panel, the quantification of the PTEN bands recognized by BA226 mAb is shown. Data are shown as relative mAb reactivity. (**D**) Recognition of PTEN by BA226 mAb (25 µg/mL) (green), as monitored by immunofluorescence on COS-7 cells transfected with pRK5 plasmid encoding human PTEN. Nuclei were stained with Hoechst. (**E**) Immunohistochemistry analysis of FFPE samples of prostate adenocarcinoma using anti-PTEN BA226 and anti-PTEN 6H2.1 mAb [a and b, positive staining; d and e, negative staining; c and f, hematoxilin-eosin staining (H/E)].

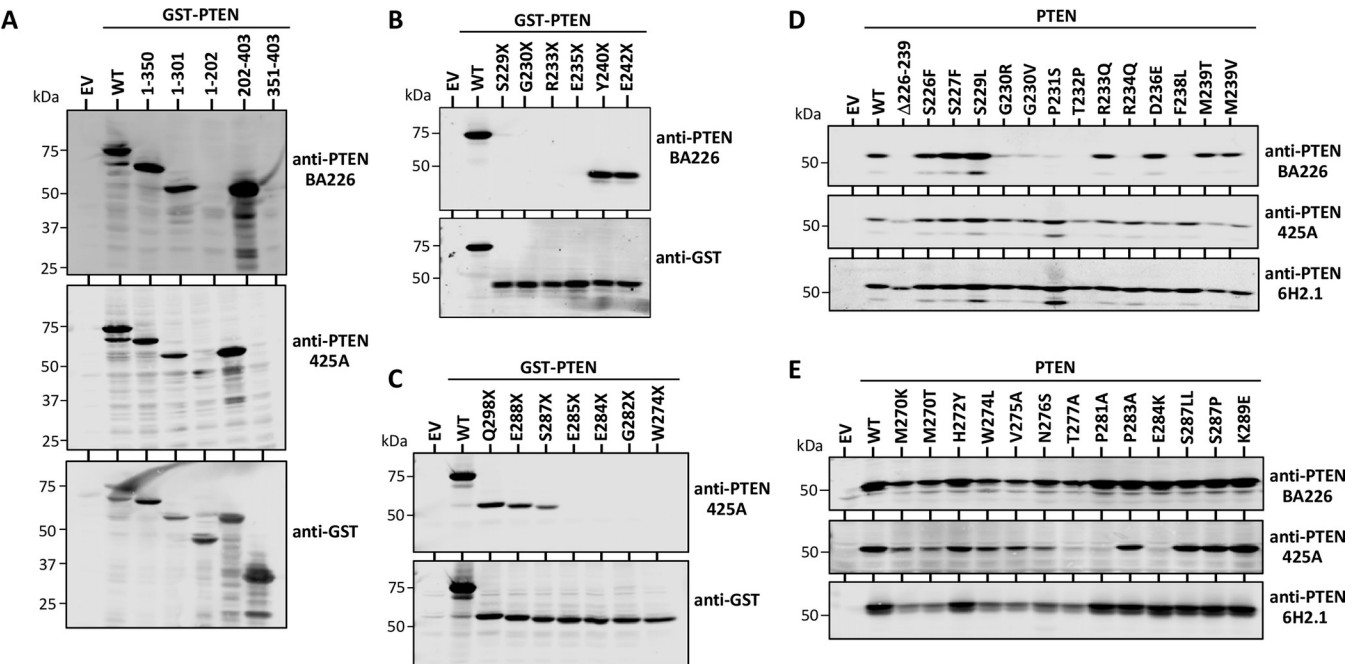

**Fig 2. Precise characterization of the epitopes recognized by anti-PTEN BA226 and 425A mAb. (A)** Immunoblot reactivity of anti-PTEN BA226 and 425A mAb with GST-PTEN fusion proteins (GST N-terminal) from lysates from COS-7 cells transfected with pRK5 empty vector (EV) or with pRK5 encoding GST-PTEN wild type (WT) or GST-PTEN truncated forms (numbers correspond to amino acid numbering). Anti-GST immunoblot is shown as a control. **(B, C)** Immunoblot reactivity of anti-PTEN BA226 (B) or anti-PTEN 425A (C) mAb with GST-PTEN fusion proteins (GST N-terminal) from lysates from COS-7 cells transfected with pRK5 empty vector (EV) or with pRK5 encoding GST-PTEN wild type (WT) or GST-PTEN C-terminal truncations generated by disease-associated nonsense mutations. Anti-GST immunoblot is shown as a control. **(D, E)** Immunoblot reactivity of anti-PTEN BA226 and anti-PTEN 425A mAb with PTEN proteins from lysates from COS-7 cells transfected with pRK5 empty vector (EV) or with pRK5 encoding PTEN wild type (WT) or PTEN variants generated by disease-associated missense mutations targeting the epitope recognized by BA226 (D) or by 425A (E) mAb. In D, a PTEN protein (Δ226–239) lacking the peptide used for mice immunization is included. Anti-PTEN 6H2.1 mAb, which recognizes an epitope at the PTEN C-terminus [34], is shown as a control. Amino acids are indicated using the single-letter code amino acid numbering. X indicates stop codon.

## Expression of PTEN isoforms and recognition by anti-PTEN C2 domain mAb

In addition to PTEN canonical isoform (403 residues), alternative translation initiation and alternative mRNA splicing generate distinct PTEN isoforms. These include the N-terminal extended isoforms PTEN-L/α (576 amino acids), PTEN-M/β (549 amino acids), and PTEN-O/ε (475 amino acids), as well as the PTEN-Δ isoform, which lacks 60 residues at the PTEN C-terminal region (**Fig 4B**). Immunoblot analysis showed that the anti-PTEN C2 domain BA226 and 425A mAb recognize all PTEN isoforms, whereas the anti-PTEN C-terminus 6H2.1 mAb does not recognize the PTEN-Δ isoform (**Fig 5A**). Recombinant PTEN-L/-M/-O starting with an artificial methionine (Met) were translated as single long PTEN isoforms, whereas the translation of PTEN-L, PTEN-M, and PTEN-O starting with their natural initiation residues [leucine (Leu), isoleucine (Ile), and leucine (Leu), respectively] rendered a mix of long and canonical PTEN isoforms, the canonical PTEN being the more abundant protein (**Fig 5A**). This is in accordance with the lower expression of PTEN-L/-M/-O proteins in human tissues and cell lines when compared to PTEN. The expression of PTEN-Δ in the background of PTEN-L/-M/-O (Met) is also shown (**Fig 5A**). To facilitate the analysis of PTEN-L/-M/-O without interference of canonical PTEN, additional experiments were performed with PTEN-L/-M/-O (Met). PTEN with C-terminal truncations caused by nonsense mutations

**Table 1. Reactivity of BA226 mAb with PTEN Ala-scanning mutations and C-terminal truncations associated to disease[1].**

| Mutation | Reactivity BA226 |
|---|---|
| S226A | + |
| S227A | + |
| N228A | + |
| S229A | + |
| G230A | + |
| P231A | +/- |
| T232A | + |
| R233A | + |
| R234A | - |
| E235A | +/- |
| D236A | +/- |
| K237A | - |
| F238A | - |
| M239A | + |
| S229X | - |
| G230X | - |
| R233X | - |
| R234X | - |
| E235X | - |
| Y240X | + |
| E242X | + |

[1]Ala substitution PTEN variants were made in the background of PTEN, and C-terminal truncations were made in the background of GST-PTEN. Variants were ectopically expressed in COS-7 cells, and mAb reactivity was monitored by immunoblot, as in Fig 1. Amino acids are indicated using the single-letter code amino acid numbering. X indicates stop codon. +, reactivity similar to PTEN WT; +/- diminished reactivity; -, lack of reactivity.

found in association with disease, including E352X and Y379X mutations, were also recognized by the anti-PTEN C2 domain BA226 and 425A mAb, but not by the anti-PTEN C-terminus 6H2.1 mAb (**Fig 5B**). Nuclear PTEN, as illustrated using the PTEN Y379X C-terminal truncation mutation that accumulates in the nucleus, was also recognized by immunofluorescence with the novel anti-PTEN BA226 mAb (**Fig 5C**).

## Functional analysis of PTEN isoforms and C-terminal truncations

Next, we tested the function of the distinct PTEN isoforms and C-terminal truncations using the phosphorylation of AKT as a surrogate marker of PTEN PIP3-phosphatase activity in cells. PTEN-Δ, both in the background of canonical PTEN or PTEN-L/-M/-O(Met) did not display PIP3-phosphatase activity, as compared with PTEN or PTEN-L/-M/-O(Met) (**Fig 6A**). However, the PTEN C-terminal truncated mutations E352X and Y379X showed similar PIP3-phosphatase activity to PTEN wild type isoforms (**Fig 6B**). A correlation was found between lack of PTEN functional activity in cells and the PTEN steady-state expression levels achieved in the experiments, suggesting that PTEN-Δ isoforms have diminished activity in cells likely by compromised protein stability. These results illustrate the existence of a variety of PTEN isoforms and disease-associated variants with distinct functional properties, whose detection under physiologic and pathogenic conditions requires the use of specific anti-PTEN mAb.

**Table 2. Reactivity of 425A mAb with PTEN Ala-scanning mutations and C-terminal truncations associated to disease[1].**

| Mutation | Reactivity 425A |
|---|---|
| S268A/K269A | + |
| M270A/F271A | + |
| H272A/F273A | + |
| W274A/V275A | +/- |
| F278A/F279A | - |
| I280A/P281A | - |
| G282A/P283A | - |
| E284A/E285A/ T286A/S287AA | - |
| E288A/K289A/ V290A/E291A | + |
| W274X | - |
| G282X | - |
| E284X | - |
| E285X | - |
| S287X | +/- |
| E288X | + |
| Q298X | + |

[1]Ala substitution PTEN variants were made in the background of PTEN, and C-terminal truncations were made in the background of GST-PTEN. Variants were ectopically expressed in COS-7 cells, and mAb reactivity was monitored by immunoblot, as in Fig 1. Amino acids are indicated using the single-letter code amino acid numbering. X indicates stop codon. +, reactivity similar to PTEN WT; +/- diminished reactivity; -, lack of reactivity.

## Discussion

The use of mAb with precise targeted specificity towards defined protein epitopes constitutes a general approach to analyse the expression and function of specific proteins. In addition, mAb

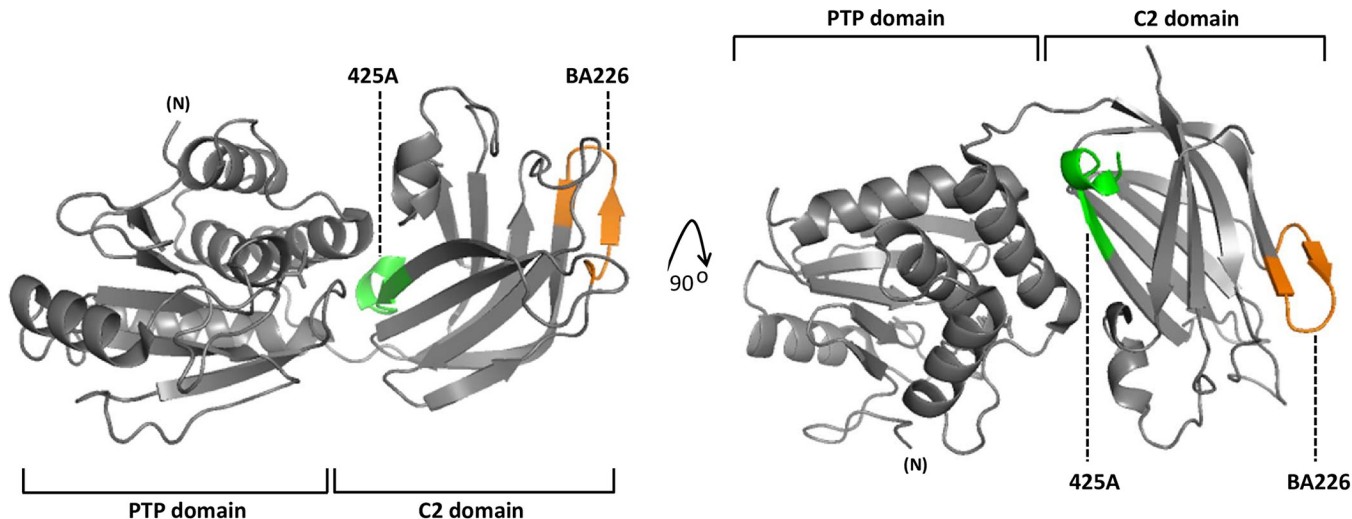

**Fig 3. Representation on PTEN 3D structure of the epitopes recognized by BA226 and 425A mAb.** PTEN PTP and C2 domains are represented according to accession ID5R [42]. BA226 epitope (residues 230–238) is in orange, and 425A epitope (residues 274–287) is in green.

**Table 3. Reactivity of BA226 mAb with PTEN disease-associated missense mutations[1].**

| Mutation | Reactivity BA226 |
|---|---|
| S226F | + |
| S227F | + |
| S229L | + |
| G230E | + |
| G230R | +/- |
| G230V | +/- |
| P231S | +/- |
| T232P | - |
| R233G | + |
| R233Q | + |
| R234G | - |
| R234L | - |
| R234Q | - |
| R234W | - |
| E235G | + |
| E235K | + |
| D236E | + |
| D236H | + |
| F238L | - |
| M239R | + |
| M239V | + |
| M239T | + |

[1]PTEN variants were made in the background of PTEN. Variants were ectopically expressed in COS-7 cells, and mAb reactivity was monitored by immunoblot, as in Fig 1. Amino acids are indicated using the single-letter code amino acid numbering. +, reactivity similar to PTEN WT; +/- diminished reactivity; -, lack of reactivity.

targeting clinically relevant proteins are the basis of many current therapeutic protocols in human disease, with a special importance in oncology [43–46]. The analysis of the expression, subcellular localization, and function of the PTEN tumor suppressor protein has an important diagnostic and prognostic potential in human cancer treatment, and several anti-PTEN mAb that recognize the PTEN C-terminus are under routine use in clinical oncology [34, 36]. In this context, the use of antibodies recognizing different PTEN regions is relevant to detect and to study alternative PTEN isoforms and proteoform variants, making relevant the generation and precise characterization of epitope-specific anti-PTEN mAb. We have obtained and precisely characterized two anti-PTEN mAb recognizing non-overlapping epitopes at the PTEN C2 domain (residues 230–238, BA226 mAb; residues 274–287, 425A mAb), and use them to study the properties of PTEN isoforms and disease-associated PTEN variants which, in some cases, are not recognized by most of the current anti-PTEN mAb. In addition, we have identified disease-associated PTEN protein variants with single amino acid substitutions which lack the BA226 or 425A epitopes, making possible the use of these mAb to monitor, in comparison with anti-PTEN mAb targeting other epitopes, the relative expression of pathogenic variants of PTEN in PHTS patients harbouring specific *PTEN* gene mutations. Distinct substitutions of specific amino acids (Arg234 in the case of BA226 mAb; Thr277 in the case of 425A mAb) abrogated the reactivity of the BA226 mAb or the 425A mAb, attributing major antigenicity properties to those residues. Although the possibility of cross-reactivity of these anti-PTEN mAb cannot be ruled out, only PTEN protein is found in the human database containing the

**Table 4. Reactivity of 425A mAb with PTEN disease-associated missense mutations[1].**

| Mutation | Reactivity 425A |
|---|---|
| M270K | + |
| M270T | + |
| H272Y | + |
| W274L | + |
| V275A | + |
| N276K | +/- |
| N276S | +/- |
| T277A | - |
| T277I | - |
| T277R | - |
| F278L | + |
| P281A | - |
| G282R | - |
| P283A | + |
| P283L | + |
| E284K | - |
| E285K | - |
| S287L | + |
| S287P | + |
| E288K | + |
| K289E | + |
| K289R | + |

[1]PTEN variants were made in the background of PTEN. Variants were ectopically expressed in COS-7 cells, and mAb reactivity was monitored by immunoblot, as in Fig 1. Amino acids are indicated using the single-letter code amino acid numbering. +, reactivity similar to PTEN WT; +/- diminished reactivity; -, lack of reactivity.

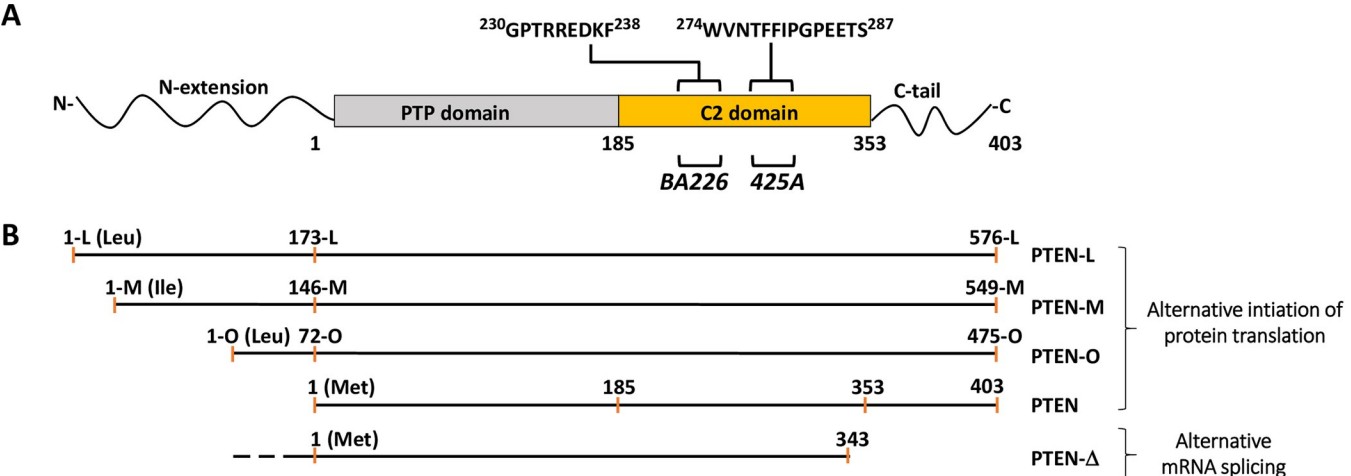

**Fig 4. Schematic representation of PTEN isoforms, with indication of BA226 and 425A epitopes.** (**A**) Depiction of PTEN domains, including the extension corresponding to the longer PTEN-L isoform. Amino acid numbering corresponds to PTEN canonical isoform (1–403). The amino acid sequences recognized by BA226 and 425A mAb are indicated using the single-letter amino acid code. (**B**) Depiction of the amino acid numbering and length of PTEN isoforms, according to [16]. The translation initiation amino acid for each isoform is indicated in brackets.

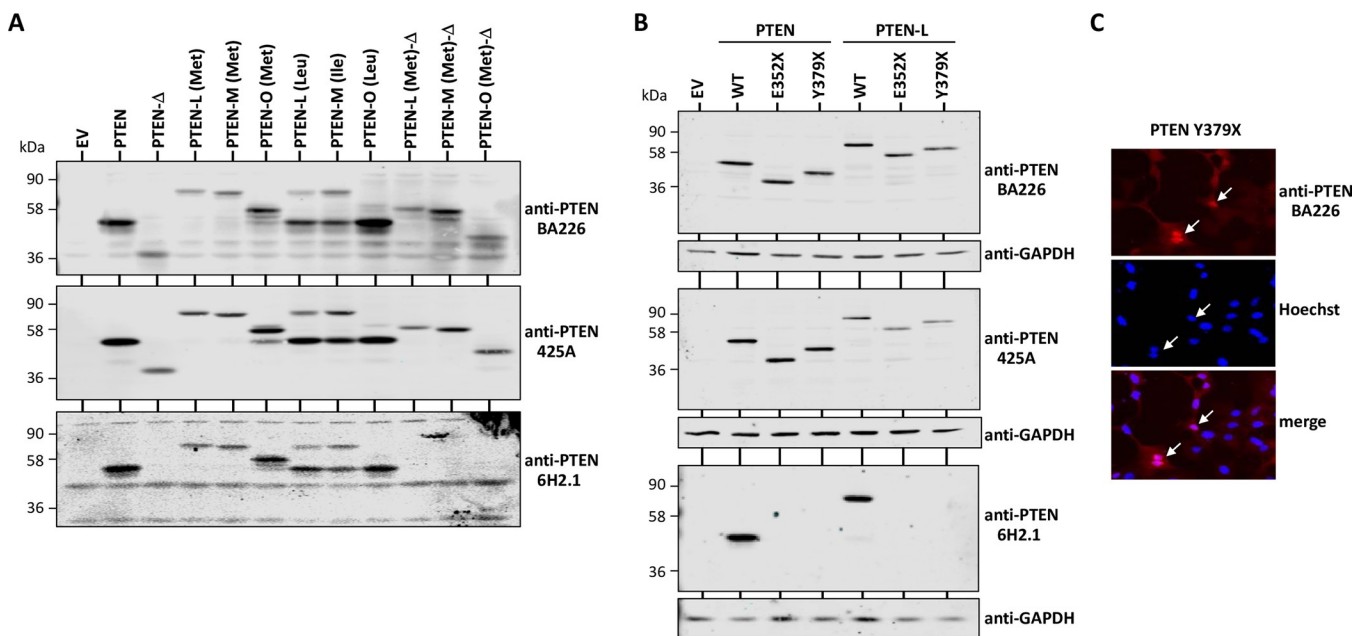

**Fig 5. Expression of PTEN isoforms and PTEN C-terminal truncations and their recognition by BA226 and 425A mAb.** (A) Cell lysates from COS-7 cells transfected with pRK5 empty vector (EV) or with pRK5 encoding the distinct PTEN isoforms were subjected to immunoblot with the indicated anti-PTEN mAb. The translation initiation amino acid for the expressed PTEN long isoforms is indicated in brackets. (B) Cell lysates from COS-7 cells transfected with pRK5 empty vector (EV) or with pRK5 encoding PTEN or PTEN-L (Met) wild type (WT) or C-terminal truncations (E352X, Y379X) generated by disease-associated nonsense mutations were subjected to immunoblot with the indicated anti-PTEN mAb. Amino acids are indicated using the single-letter code amino acid numbering. (C) Recognition of nuclear PTEN by BA226 mAb (red), as monitored by immunofluorescence on COS-7 cells transfected with pRK5 plasmid encoding PTEN Y379X. Arrows indicate PTEN Y379X positive cells. Nuclei were stained with Hoechst. In B and C, amino acids are indicated using the single-letter code amino acid numbering. X indicates stop codon.

peptides encompassing PTEN residues 230–238 or 274–287. The PTEN amino acid sequences recognized by the anti-PTEN BA226 and 425A mAb are conserved between mammalian species, but they are not conserved in several vertebrate non-mammalian model organisms. For instance, the BA226 epitope is weakly conserved in clawed frog (*Xenopus tropicalis*) and zebra-fish (*Danio rerio*), whereas the 425A epitope is weakly conserved in chicken (*Gallus gallus*). This potentially precludes the use of the anti-PTEN BA226 and 425A mAb to study PTEN in particular non-mammalian species, but makes these reagents suitable to specifically monitor the heterologous expression of human PTEN variants in non-mammalian experimental settings, without interference with the detection of the non-mammalian host PTEN.

PTEN protein truncations generated by nonsense mutations are unstable proteins with impaired function, although those generated by nonsense mutations targeting the PTEN C-terminal tail retain most of their stability and PIP3 phosphatase activity in cells [12, 13]. The BA226 epitope contains Arg233, which is one of the PTEN residues frequently targeted by nonsense mutations in human tumors and in the germline of PHTS patients, and it constitutes a suitable mAb to study PTEN C-terminal truncations downstream PTEN residue 239. In our experiments, PTEN C-terminal truncated proteins generated by the nonsense mutations E352X and Y379X displayed steady-state expression levels and PIP3 phosphatase activity at similar levels to PTEN wild type. On the other hand, the PTEN-Δ isoform, whose truncation is at residue 343 at the end of the C2 domain, was expressed with lower abundance and displayed impaired PIP3 phosphatase activity in cells. This was observed on the background of PTEN and PTEN long isoforms. It has been proposed a similar function for PTEN-Δ and canonical PTEN in renal cancer cells [31], although to allow PTEN-Δ detection it was used in the

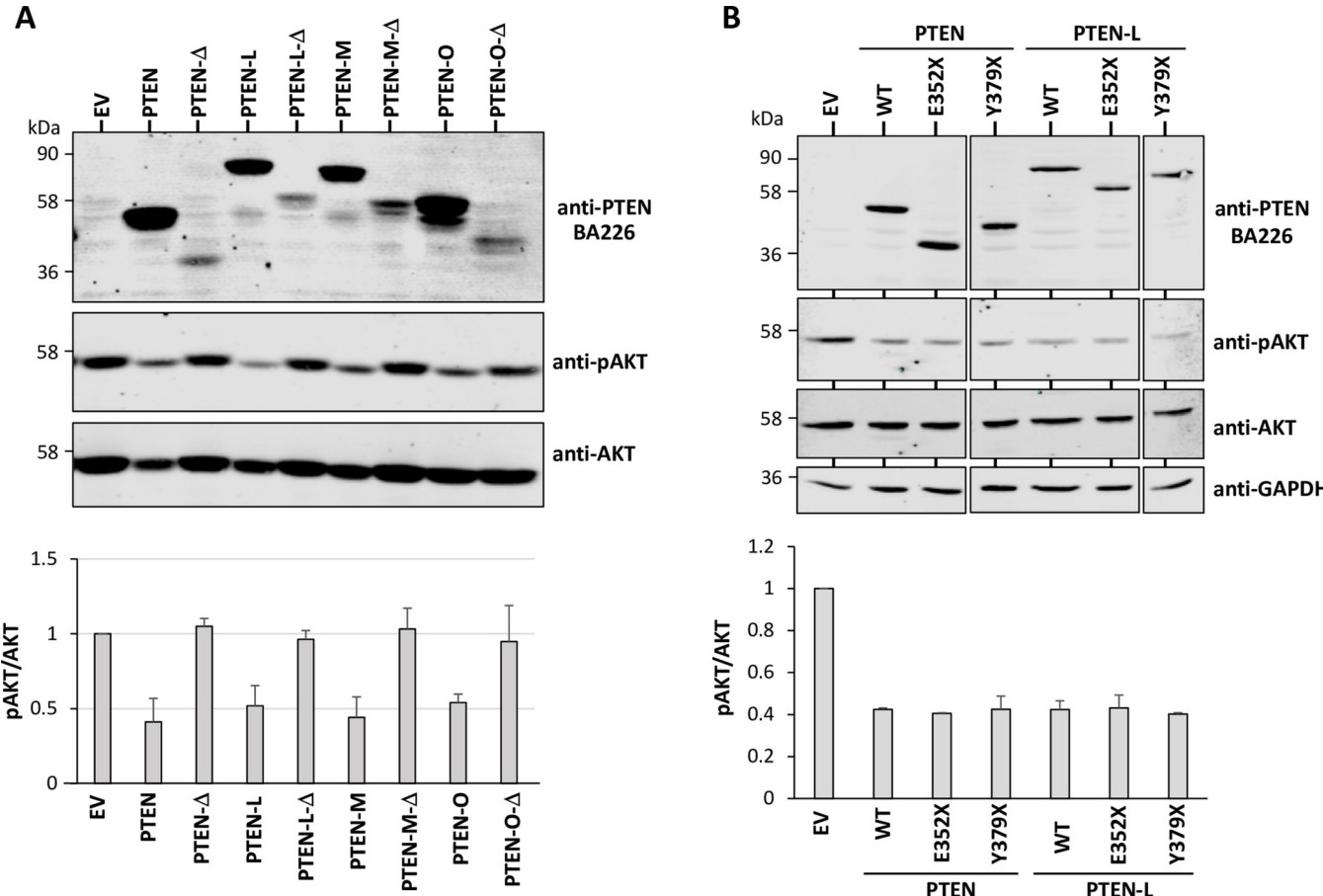

**Fig 6. Functional activity of PTEN isoforms and PTEN C-terminal truncations.** (A) COS-7 cells were transfected with pRK5 empty vector (EV) or pRK5 encoding the distinct PTEN isoforms [PTEN, PTEN-Δ, PTEN-L/-M/-O(Met)], in combination with the pSG5 HA-AKT1 plasmid, and cells were processed for immunoblot using anti-PTEN BA226, anti-pAKT, or anti-AKT antibodies, as indicated. PTEN long isoforms correspond to PTEN-L/-M/-O(Met). Top panel shows a blot from a representative experiment, and bottom panel shows the pAKT/AKT ratio from each PTEN isoform, after quantification of the bands from at least two independent experiments. Differences in the activity of PTEN vs PTEN-Δ, PTEN-L vs PTEN-L-Δ, and PTEN-M vs PTEN-M-Δ were statistically significant (p<0.05). (B) COS-7 cells were transfected with pRK5 empty vector (EV) or pRK5 encoding PTEN C-terminal truncations generated by disease-associated nonsense mutations, in combination with the pSG5 HA-AKT1 plasmid, and cells were processed as in A. Top panel shows a blot from a representative experiment, and bottom panel shows the pAKT/AKT ratio from each PTEN variant, after quantification of the bands from at least two independent experiments. No significant differences in the activity of PTEN or PTEN-L vs the respective truncation mutants were observed. Amino acids are indicated using the single-letter code amino acid numbering. X indicates stop codon.

experimental setting PTEN-Δ C-terminally tagged, which could interfere with PTEN-Δ loss of stability. Caspase-3 cleavage of PTEN C-terminal tail generates PTEN C-terminal truncated proteoforms with potential physiologic activity but altered functional regulation [47], whose monitoring using specific anti-PTEN mAb is also of interest.

PHTS patients harbour one *PTEN* wild type allele and, in many cases, a *PTEN* mutated allele encoding a PTEN protein with variable stability and function. We speculate that the complexity in the expression of PTEN proteoform variants under distinct pathogenic conditions could be responsible, at least in part, of the wide phenotypic diversity found in PHTS patients. However, how the relative expression and function of the patient-specific PTEN proteoform variants is associated with pathogeny is not regularly monitored, which is mainly due to the lack of variant-specific anti-PTEN mAb. The generation and precise characterization of anti-PTEN mAb against distinct PTEN epitopes, as the ones here described, will be highly valuable to understand PHTS pathogeny associated to specific PHTS *PTEN* gene mutations.

## Supporting information

**S1 Raw images.**
(PDF)

## Acknowledgments

We thank all personnel from the Genetics-Genomics Core facility, Biobizkaia Health Research Institute, for their expert assistance with DNA sequencing; and Abyntek for their advice and assistance.

## Author Contributions

**Conceptualization:** Rafael Pulido.

**Funding acquisition:** José I. López, Rafael Pulido.

**Investigation:** Leire Torices, Caroline E. Nunes-Xavier, José I. López, Rafael Pulido.

**Methodology:** Leire Torices, Caroline E. Nunes-Xavier, Rafael Pulido.

**Supervision:** Rafael Pulido.

**Visualization:** Leire Torices, Caroline E. Nunes-Xavier, José I. López.

**Writing – original draft:** Rafael Pulido.

**Writing – review & editing:** Leire Torices, Caroline E. Nunes-Xavier, José I. López, Rafael Pulido.

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
