## [Decision Letter · Decision Letter 0]

4 May 2023

PONE-D-23-09497Novel anti-PTEN C2 domain mAb to analyse the expression and function of PTEN isoform variantsPLOS ONE

Dear Dr. Pulido,

Thank you for submitting your manuscript to PLOS ONE. After careful consideration, we feel that it has merit but does not fully meet PLOS ONE’s publication criteria as it currently stands. Therefore, we invite you to submit a revised version of the manuscript that addresses the points raised during the review process.

We look forward to receiving your revised manuscript.

Kind regards,

Avaniyapuram Kannan Murugan, M.Phil., Ph.D.

Academic Editor

PLOS ONE

Journal Requirements:

   "This work has been supported in part by grants BBH-19-001 (to RP) from PTEN Research Foundation (United Kingdom), and SAF2016-79847-R (to RP and JIL), from Ministerio de Economía y Competitividad (Spain and The European Regional Development Fund). LT has been the recipient of a predoctoral fellowship from Asociación Española Contra el Cáncer (AECC, Junta Provincial de Bizkaia, Spain). CN-X is the recipient of a Miguel Servet Research Contract from Instituto de Salud Carlos III (grant number CP20/00008). "

   "This work has been supported in part by grants BBH-19-001 (to RP) from PTEN Research Foundation (United Kingdom), and SAF2016-79847-R (to RP and JIL), from Ministerio de Economía y Competitividad (Spain and The European Regional Development Fund). LT has been the recipient of a predoctoral fellowship from Asociación Española Contra el Cáncer (AECC, Junta Provincial de Bizkaia, Spain). CN-X is the recipient of a Miguel Servet Research Contract from 

Instituto de Salud Carlos III (grant number CP20/00008). We thank all personnel from the Genetics-Genomics Core facility, Biocruces Bizkaia Health Research Institute, for their expert assistance with DNA sequencing; and Abyntek for their advice and assistance."

  "This work has been supported in part by grants BBH-19-001 (to RP) from PTEN Research Foundation (United Kingdom), and SAF2016-79847-R (to RP and JIL), from Ministerio de Economía y Competitividad (Spain and The European Regional Development Fund). LT has been the recipient of a predoctoral fellowship from Asociación Española Contra el Cáncer (AECC, Junta Provincial de Bizkaia, Spain). CN-X is the recipient of a Miguel Servet Research Contract from Instituto de Salud Carlos III (grant number CP20/00008). "

Additional Editor Comments:

The manuscript is very interesting and the same is reflected from the reviewers however the critiques raised by the reviewers to be addressed before considering it for publication.

Reviewers' comments:

Reviewer's Responses to Questions

**Comments to the Author**

1. Is the manuscript technically sound, and do the data support the conclusions?

Reviewer #1: Yes

Reviewer #2: Yes

Reviewer #3: Partly

Reviewer #4: Yes

2. Has the statistical analysis been performed appropriately and rigorously? 

Reviewer #1: Yes

Reviewer #2: N/A

Reviewer #3: N/A

Reviewer #4: N/A

3. Have the authors made all data underlying the findings in their manuscript fully available?

Reviewer #1: Yes

Reviewer #2: Yes

Reviewer #3: Yes

Reviewer #4: Yes

4. Is the manuscript presented in an intelligible fashion and written in standard English?

Reviewer #1: Yes

Reviewer #2: Yes

Reviewer #3: Yes

Reviewer #4: Yes

5. Review Comments to the Author

Reviewer #1: In this manuscript, Torices et al., describe the generation and full characterization of a new antibody against the PTEN tumor suppressor. While the generation of monoclonal antibodies in general could not represent a sufficient relevant subject for a manuscript in PLOS ONE, it is important to take into account the target. PTEN gene is inactivated in a variety of human cancers and in relevant cancer predisposition syndromes, moreover its inactivation could be extremely relevant for the predictive response of the tumors to particular therapies. In this context, the assessment of potential status is a challenge. This could be solved by different genomic approaches, but the availability of a more simple and accurate method is a current need. In addition, the use of several polyclonal and mAbs against PTEN is hampered by the lack of sufficient information on the targeted epitope. Accordingly I consider that the present work merits publication in PLOS One. Nonetheless, I think there are some minor aspects that could enhance its relevance.

First of all, I would recommend expand the analysis of human samples (as those shown in Fig 1E) by immunohistochemistry including tumors (prostate is a good example) of known status of PTEN gene (different mutations, deletions, etc.) using the BA226 mAb. This could maximize the use of the antibody in the clinical practice.

In the same line it would be worth performing immunofluorescence analysis of endogenous PTEN using different cell lines, again with different known PTEN status. Is this antibody reacting similarly with the different localizations of endogenous PTEN (mainly nuclear and cytoplasm)? Given that the functional roles of PTEN may differ between these two localizations this aspect could also be relevant

One potential interesting aspect is to know whether this antibody reacts also with other species.

Reviewer #2: Pulido et al. previously generated monoclonal antibodies (425A) recognizing C2 domain of PTEN protein. Here they report on the characterization of another clone, BA226, recognizing different epitope in the C2 domain of PTEN. Since the antibodies previously used to detect PTEN were against its C-terminal tail region which could be lost by disease-associated nonsense mutations, antibodies recognizing other regions of PTEN are of value.

I find the work important, experiments well performed, and the manuscript well written. I only have a few minor points that I would like the authors to consider before publication.

1) The term “mAb” (in plural) is used without definition. Please define it.

2) The symbol “X” (as in “S229X”) is used without definition. Please define it.

3) Page 2, line 8: The phrase “a bias in the of analysis” does not make sense.

4) Page 12, line 11, “as well”: Should this be “as well as”?

5) Page 12, line 22, “PTEN C-terminal - - -, where also recognized ---”: This sentence does not make sense to me. If the “where” is a typo of “were”, then the sentence makes sense but is logically incorrect, since what were recognized were the proteins and the not mutations. I would therefore put it like, “PTEN with C-terminal truncations ---”.

6) Although I am not a native English speaker, usage of the following words/phrases sounds strange to me.

i) The “more” on page 3 (line 19) and page 14 (lines 3 and 24) would raise a question “more than what?”. I would use “most” (or try to eliminate “more” when appropriate).

ii) “similar than” (page 13, line 9; page 14, line 28): “similar to” sounds better to me.

Reviewer #3: The manuscript titled "Novel anti-PTEN C2 domain mAb to analyze the expression and function of PTEN isoform variants" aims to describe the generation and precise characterization of an anti-PTEN monoclonal antibody that recognizes the PTEN C2-domain. This antibody is used to monitor the expression and function of PTEN isoforms, as well as missense and nonsense mutations associated with disease. The author's focus is on finding a new mAb that recognizes PTEN C-terminal truncations which retain stability and function but have lost their epitopes.

The work shows promising applications, supported by numerous experiments; however, there are several gaps preventing publication in its current form:

Major concerns:

1. Figure 1D: Was plasmid transfection efficiency 100%? How long after plasmid transfection were cells obtained for IF experiment? Could authors add negative control?

2. Figure 1E: Could authors add commercial PTEN mAb as a positive control? Authors should include sample information (e.g., sample type).

3. Figure 2: The principle of truncated forms of PTEN should be described in the manuscript.

4. A Δ226-239 PTEN protein was performed in figure 2D, but Δ270-289 is lacking in figure 2E. Could authors explain this?

5. Please add a statistical method for calculating reactivity of BA226 and 425A.

6. Figure 5B: Please add anti-GAPDH.

Minor concerns:

1. Figure 1C: Could authors add concentration of protein?

Reviewer #4: This is a well written paper (with only one typo, line 7 of abstract which requires deletion of redundant text “of”) describing the justification and the process follow in creating two antibodies that recognising two sites on the PTEN molecule one of which is not detectable by most agents currently available. With 6 figures and 4 tables, the authors demonstrate the process undertaken to prove the specificity, though do not test an array of other non-related controls to rule out the problem of cross reactivity. Also they do not identify what clinical or research question will immediately benefit from understanding of the complexity they have identified

6. PLOS authors have the option to publish the peer review history of their article (what does this mean?). If published, this will include your full peer review and any attached files.

Reviewer #1: **Yes**

Reviewer #2: No

Reviewer #3: No

Reviewer #4: No

---

## [Decision Letter · Decision Letter 1]

18 Jul 2023

Novel anti-PTEN C2 domain monoclonal antibodies to analyse the expression and function of PTEN isoform variants

PONE-D-23-09497R1

Dear Dr. Pulido,

We’re pleased to inform you that your manuscript has been judged scientifically suitable for publication and will be formally accepted for publication once it meets all outstanding technical requirements.

Kind regards,

Avaniyapuram Kannan Murugan, M.Phil., Ph.D.

Academic Editor

PLOS ONE

Additional Editor Comments (optional):

Reviewers' comments:

Reviewer's Responses to Questions

**Comments to the Author**

1. If the authors have adequately addressed your comments raised in a previous round of review and you feel that this manuscript is now acceptable for publication, you may indicate that here to bypass the “Comments to the Author” section, enter your conflict of interest statement in the “Confidential to Editor” section, and submit your "Accept" recommendation.

Reviewer #1: All comments have been addressed

Reviewer #2: All comments have been addressed

Reviewer #3: (No Response)

2. Is the manuscript technically sound, and do the data support the conclusions?

Reviewer #1: (No Response)

Reviewer #2: Yes

Reviewer #3: Partly

3. Has the statistical analysis been performed appropriately and rigorously? 

Reviewer #1: (No Response)

Reviewer #2: Yes

Reviewer #3: N/A

4. Have the authors made all data underlying the findings in their manuscript fully available?

Reviewer #1: (No Response)

Reviewer #2: Yes

Reviewer #3: Yes

5. Is the manuscript presented in an intelligible fashion and written in standard English?

Reviewer #1: (No Response)

Reviewer #2: Yes

Reviewer #3: Yes

6. Review Comments to the Author

Reviewer #1: (No Response)

Reviewer #2: (No Response)

Reviewer #3: (No Response)

7. PLOS authors have the option to publish the peer review history of their article (what does this mean?). If published, this will include your full peer review and any attached files.

Reviewer #1: **Yes**

Reviewer #2: No

Reviewer #3: No

---

## [Editor Report · Acceptance letter]

24 Jul 2023

PONE-D-23-09497R1 

Novel anti-PTEN C2 domain monoclonal antibodies to analyse the expression and function of PTEN isoform variants 

Dear Dr. Pulido:

I'm pleased to inform you that your manuscript has been deemed suitable for publication in PLOS ONE. Congratulations! Your manuscript is now with our production department. 

Kind regards, 

on behalf of

Dr. Avaniyapuram Kannan Murugan 

Academic Editor

PLOS ONE